# SARS-CoV-2 Spike and Neutralizing Antibody Kinetics 90 Days after Three Doses of BNT162b2 mRNA COVID-19 Vaccine in Singapore

**DOI:** 10.3390/vaccines10020331

**Published:** 2022-02-18

**Authors:** Chin Shern Lau, Soon Kieng Phua, Ya Li Liang, May Lin Helen Oh, Tar Choon Aw

**Affiliations:** 1Department of Laboratory Medicine, Changi General Hospital, Singapore 529889, Singapore; soon_kieng_phua@cgh.com.sg (S.K.P.); yali_liang@cgh.com.sg (Y.L.L.); tarchoon@gmail.com (T.C.A.); 2Department of Infectious Diseases, Changi General Hospital, Singapore 529889, Singapore; helen.oh.m.l@singhealth.com.sg; 3Department of Medicine, National University of Singapore, Singapore 119077, Singapore; 4Academic Pathology Program, Duke-NUS Medical School, Singapore 169857, Singapore

**Keywords:** SARS-CoV-2, booster vaccination, antibodies

## Abstract

Background: We evaluated the post-booster (BNT162b2) antibody responses in Singapore. Methods: Participants (*n* = 43) were tested pre-booster and 20/30/60/90 days post-booster. Participants were boosted 120–240 days (mean 214 days) after their second dose and had no history or serologic evidence of prior COVID-19 infection; all participants had undetectable SARS-CoV-2 nucleocapsid antibodies throughout the study. Total nucleocapsid and spike antibodies (S-Ab) were assessed on the Roche Elecsys e802 and neutralizing antibody (N-Ab) on the Snibe quantitative N-Ab assay. Results: Pre-booster median S-Ab/N-Ab titers were 829 BAU/mL/0.83 µg/mL; 2 participants were below manufacturer’s N-Ab cut-offs of 0.3 µg/mL (0.192 and 0.229). Both S-Ab and N-Ab titers peaked at 30 days post-booster (median S-Ab 25,220 BAU/mL and N-Ab 30.3 µg/mL) at 30–37× pre-booster median levels. These peak post-booster S-Ab/N-Ab titers were 11× (25,220 vs. 2235 BAU/mL) and 9× (30.3 vs. 3.52 µg/mL) higher than the previously reported peak post-second dose levels. Antibody titers declined to 12,315 BAU/mL (51% decrease) and 14.3 µg/mL (53% decrease) 90 days post-booster. Non-linear regression estimates for S-Ab/N-Ab half-lives were 44/58 days. At 180 days post-booster, S-Ab/N-Ab are estimated to be 2671 BAU/mL/4.83 µg/mL. Conclusions: Both S-Ab and N-Ab show a good response following post-booster vaccination, with half-lives that may provide a prolonged antibody response.

## 1. Background

As the COVID-19 pandemic continues into its third year, numerous countries have started to recommend the use of a booster (third) vaccine dose to their populations to curb the pandemic. Indeed, the US Food and Drug Administration, Centers for Disease Control and Prevention began recommending a third booster dose in vaccinated individuals since September 2021 [1]. Several studies have demonstrated a brisk antibody response to the booster vaccination [2,3,4,5], with some titers even higher than post-infection levels [3]. Booster vaccination has been shown to be effective against several different SARS-CoV-2 variants, including the new Omicron variant [6,7,8], and tolerated well with minimal side effects [9]. Indeed, booster vaccination regimens have been beneficial in controlling the spread of SARS-CoV-2, with real-world analysis [10] showing that it can reduce the rates of COVID-19 by a factor of 11.3, and severe illness by a factor of 19.5.

However, booster vaccinations have yet to become widespread in southeast Asia. Thankfully, booster doses of the Pfizer BNT162b2 mRNA COVID-19 vaccine have been encouraged in our country since September 2021, initially recommended for those over 60 years old, but now rolled out for the general population. There is a paucity of data on post-booster antibody response in Asian populations. We thus undertook to evaluate the post-booster antibody responses in Singapore and compared it to previously documented spike (S-Ab) and neutralizing antibody (N-Ab) responses after the second dose of vaccine.

## 2. Methods

### 2.1. Study Participants

Participants (*n* = 43) were volunteers who agreed to donate blood samples at five different time points: pre-booster, 20/30/60/90 days post-booster between September 2021 and January 2022. All participants were fully vaccinated with a second dose of Pfizer vaccine 120 to 240 days (mean = 214 days) prior to booster with the Pfizer BioNTech vaccine and had no prior history of COVID-19 infection or exposure to COVID-19, as evidenced by negative nucleocapsid antibodies. All participants remained COVID-19 naïve throughout the study. Due to different vaccination schedules, the number of samples at each time point was different. The subjects ranged in age from 22–73 years (mean 48.0 ± 14.7 years), with 32.6% males (14/43) and 67.4% females (29/43).

### 2.2. Analytical Methods

Serum at each time point was obtained and stored at −70 degrees Celsius if not immediately analysed. Frozen samples were thawed for 1 h at room temperature just prior to analysis. Thawed samples were vortexed before analysis. The Roche Elecsys Anti-SARS-CoV-2 S quantitative double-antigen sandwich electro-chemiluminescent immunoassay (run on the Roche Elecsys e801 auto-analyzer) and the Snibe competitive quantitative N-Ab assay (run on the Snibe Maglumi) have been previously evaluated and described by our laboratory [11,12]. The Snibe N-Ab assay reports a clinical sensitivity of 100% (95% CI 93.69–100) in 57 samples with confirmed SARS-CoV-2 VNT_50_ ≥ 20 in their manufacturer’s insert. The assay has been compared to plaque reduction neutralization tests [13]. Nucleocapsid antibodies were determined on the Roche Anti-SARS-CoV-2 nucleocapsid antibody assay as previously described [14]. S-Ab results can be converted to WHO international units based on a user circular provided by the manufacturer (BAU/mL = 0.97 × U/mL). As specified by the manufacturer, samples are considered S-Ab positive at ≥0.8 U/mL (0.78 BAU/mL), with ≥0.3 μg/mL regarded as positive for N-Ab.

### 2.3. Statistical Analysis

Data were presented as the median (interquartile range) where appropriate. No indeterminate or missing results were used. Standard regression analysis was also performed to assess the agreement between S-Ab and N-Ab titers using MedCalc Statistical Software (version 20.008, MedCalc Software Ltd., Ostend, Belgium). To assess the antibody half-life post-vaccinations, we utilized a simple non-linear regression model that correlated the log_10_ antibody levels to days postvaccination using GraphPad Prism (GraphPad Prism, version 9.2.0, GraphPad Software, San Diego, CA, USA). Our institution’s institutional review board deemed this work exempt as this was part of a seroprevalence survey. However, informed consent was obtained from all subjects involved in the study, as they needed to provide blood samples on several occasions. The study was conducted in compliance with STARD guidelines (see Appendix A).

## 3. Results

### 3.1. Total S-Ab vs. N-Ab Responses after Booster Vaccination

Median pre-booster antibody levels were S-Ab 829 BAU/mL and N-Ab 0.83 µg/mL. All participants were seropositive for total S-Ab and N-Ab prior to the third dose of vaccine except for 2 N-Ab seronegative participants (0.192 and 0.229 µg/mL). Both S-Ab and N-Ab titers peaked at 30 days post-booster dose in all subjects; peak S-Ab was 25,220 BAU/mL and peak N-Ab was 30.3 µg/mL (see Figure 1). This represents a 30× and 37× increase in antibody titers from pre-booster levels. S-Ab and N-Ab titers declined thereafter, to 12,315 BAU/mL (51% decrease) and 14.3 µg/mL (53% decrease) by 90 days post-booster. We had two cases with total S-Ab testing at all time points, with antibody kinetics demonstrated in Appendix A.

### 3.2. Regression Analysis

Regression analysis of all cases after the second and third vaccine doses showed close agreement between S-Ab and N-Ab (Pearson correlation, r = 0.95) (see Figure 2).

### 3.3. Comparison between Post-Second Dose and Post-Booster Responses

We compared the post-booster antibody responses to the post-second vaccine dose responses from our previous data set [12]. The peak post-booster S-Ab titer was 11-fold (25,220 vs. 2235 BAU/mL) higher than the peak antibody response after the second dose of the Pfizer vaccine while N-Ab was 9-fold higher (30.3 vs. 3.52 µg/mL) (see Figure 3).

### 3.4. Antibody Half-Life Analysis

From non-linear regression decay curves for S-Ab and N-Ab, the half-lives for S-Ab and N-Ab were 44 days and 58 days, respectively (see Figure 4). It would thus take an estimated 255/326 days for S-Ab/N-Ab to decrease from peak titers to original pre-booster levels. At 6 months (180 days) post-booster, the projected levels of S-Ab and N-Ab would be 2671 BAU/mL and 4.83 µg/mL, still well above pre-booster levels of 829 BAU/mL and 0.83 µg/mL, respectively. We also calculated the half-life for total S-Ab and N-Ab from 20–90 days post-second vaccination from our prior data set [12], and total S-Ab had a half-life of 62 days, but N-Ab had a half-life of 48 days (see Appendix A).

## 4. Discussion

Our study supports the notion that antibody responses post-booster vaccination in our population are robust. The peak antibody titers post-mRNA booster vaccination were 30/37× (S-Ab/N-Ab) higher than pre-booster levels, and 11/9× higher than peak levels after the second inoculation. These findings are similar to those reported in other countries [2,3,4,5]. We also confirm the further decline of both S-Ab and N-Ab beyond 180 days after the second dose of mRNA vaccine, extending the findings of our previous study [12]. The S-Ab/N-Ab titers dropped by 63%/76% between the peak post-second dose levels and pre-booster levels. This is supported by another study [15] where Roche total S-Ab decreased by 37% and 57% 3 and 6 months after the peak post-second dose.

A precise protective antibody titer against SARS-CoV-2 infection is not known, although some studies have correlated antibody titers with neutralization levels [16] and vaccine efficacy (one minus risk-ratio times 100) [17]. A recent preprint [18] suggested a reduced risk of infection post vaccination with Roche total S-Ab levels >485 BAU/mL (500 U/mL). Notably, some of their patients had breakthrough infections with total S-Ab titers of 906 BAU/mL (934 U/mL). Vaccine protection is complicated by the presence of newer SARS-CoV-2 variants. However, there is promising evidence that booster vaccines are effective in preventing infection from both Omicron and Delta variants [19]; moreover, the viral loads are lower (higher RT-PCR cycle thresholds) in post-booster cases than in cases with just two vaccinations [20]. Furthermore, some studies have already shown that rates of confirmed COVID-19 and severe illness were substantially lower among those who received a booster (third) dose of the BNT162b2 vaccine [10], and that booster vaccinations significantly reduced the incidence of SARS-CoV-2 infections (116 per 100,000 person-days prior to booster vaccination versus 12.8 per 100,000 after booster vaccination) for an estimated relative reduction of 93% (hazard ratio 0.07) [21]. Thus, the robust antibody response post-booster vaccination in our study supports the real-world benefit in blunting the pandemic’s progress.

One novel finding from our study is that we have demonstrated a robust antibody response at 90 days after the booster vaccination and derived theoretical half-lives for S-Ab and N-Ab after the booster vaccination. The shorter half-life of total S-Ab post-booster may be attributed to faster-declining antibodies (e.g., IgM), however, N-Abs do have a longer half-life post-booster compared to post-second dose (58 vs. 48 days). Although further studies are required to fully demonstrate the antibody decay after the booster dose at >180 days later, the half-life of both antibodies post-booster is still robust (44 days for S-Ab, 58 days for N-Ab), and are estimated to return to pre-booster levels after 8–11 months. Indeed, some estimate that the antibody responses after natural infection/vaccination may last for only 1–2 years [22], with other studies [23] demonstrate that at 180 days after the second BNT162b2 vaccination S-Ab titers were similar to those in persons vaccinated with only one dose of vaccine, or in COVID-19-convalescent individuals. Thus, booster doses would be important to maintain a prolonged antibody response beyond the period of raised antibody titers >180 days after two doses of vaccine, even if some waning is demonstrated by 90 days post-booster. In one large study of boosted subjects [24], some waning in protection against symptomatic COVID-19 at 10 weeks (70 days) post-booster was seen but protection against severe disease was maintained.

Another strength of our study is that we report specific Roche total S-Ab/Snibe N-Ab post-booster responses from a healthy COVID-19-naïve population, who all adhered to a regimen of three doses of BNT162b2 vaccine. However, in our country, booster doses were only approved from September 2021. As such, this led to the relatively longer interval between the second and third dose in our vaccination regimen, with only three subjects receiving booster doses <180 days after their second vaccination (mean interval of 214 days in our population). The interval between vaccine doses may affect the resulting antibody response. Some studies recommend a longer interval to establish a more durable response and more time to develop long-term immunity [25]. In one study [5], geometric mean responses were compared between groups of patients who were boosted at 186 versus 262 days post-second dose (Comirnaty vaccine); peak post-booster antibody geometric mean responses 2–4 weeks post-booster were higher in the extended dosing interval group (geometric mean titers 13,980 versus 18,104). Thus, the longer dosing interval may have been a factor that contributed somewhat to the high peak post-booster antibody levels we found. However, the dilemma in trying to decide upon a suitable dosing interval would have to be balanced between declining antibody titers after the second dose of vaccine and the desired peak post-booster antibody levels. Further studies and discussions among the scientific community would be required to decide upon the optimal dosing interval for boosters.

A limitation of our study is that we were only able to assess the response to one vaccine type (three doses of BNT162b2) and our results may not be generalizable to patients who receive different types of vaccine or heterologous combinations of vaccines [2]. The populations we studied for the post-second dose and post-booster dose responses were not the same individuals. We do not yet have data after 90 days post-booster. Furthermore, as SARS-CoV-2 antibody tests have great variation between each other [26], our results may also not be generalizable to other antibody assays.

## 5. Conclusions

We confirm that the S-Ab and N-Ab antibody responses post-booster vaccination are robust in our study population. There is good agreement between post-vaccination S-Ab and N-Ab. Both S-Ab and N-Ab show a good response post-booster vaccination, with half-lives that are expected to provide a more prolonged antibody response.

## Figures and Tables

**Figure 1 vaccines-10-00331-f001:**
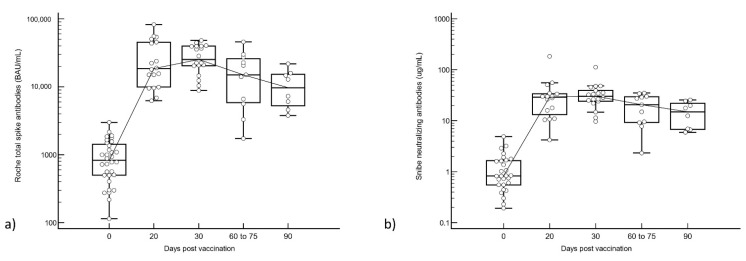
(**a**) Roche total spike and (**b**) Snibe neutralizing antibodies pre-booster and after booster vaccination. Antibodies are displayed on a semi-logarithmic scale.

**Figure 2 vaccines-10-00331-f002:**
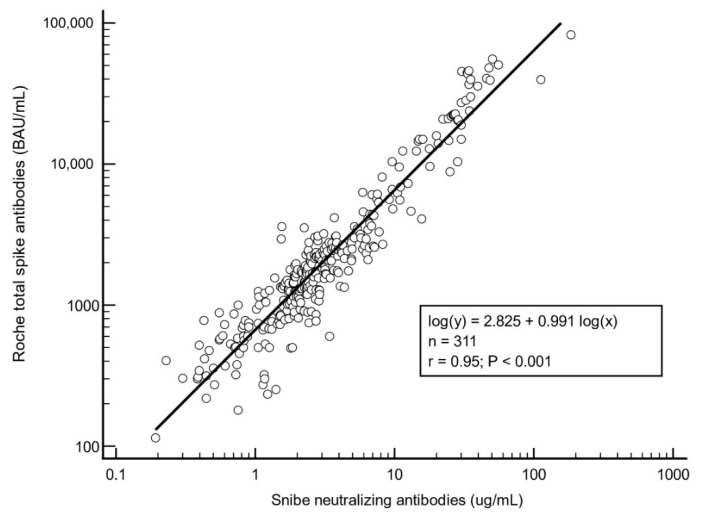
Regression analysis between total spike and neutralizing antibodies after second and third vaccinations. Data used in this figure includes data from this study (current post-booster population) and from our previous study (prior post-second dose population) [12] and are thus different individuals.

**Figure 3 vaccines-10-00331-f003:**
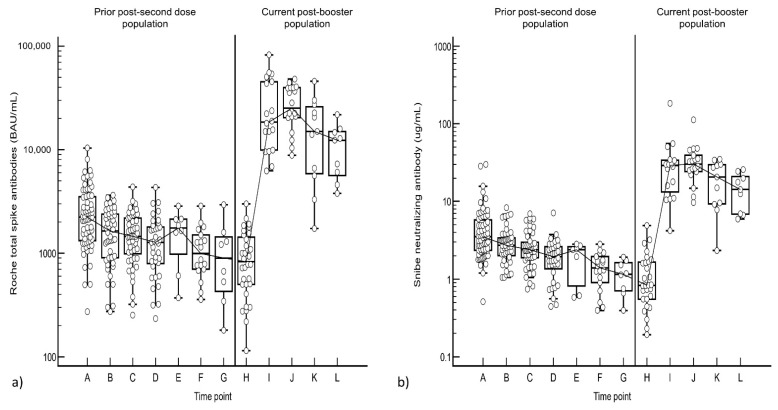
(**a**) Total spike and (**b**) Neutralizing Antibody trend between the second and third vaccination doses. The time points are: A: D2D20, B: D2D40, C: D2D60, D: D2D90, E: D2D120, F: D2D150, G: D2D180, H: Pre-D3, I: D3D20, J: D3D30, K: D3D60-75, L: D3D90. Data used in this figure includes data from this study (current post-booster population) and from our previous study (prior post-second dose population) [12] and are thus different individuals.

**Figure 4 vaccines-10-00331-f004:**
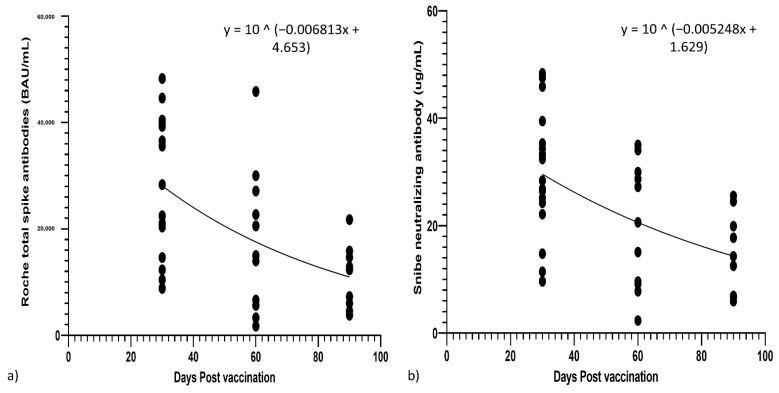
Non linear regression of (**a**) Roche total spike and (**b**) Snibe neutralizing antibodies from their peak at 30 days post-booster vaccination.

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
