# Peer review of "SARS-CoV-2 Spike and Neutralizing Antibody Kinetics 90 Days after Three Doses of BNT162b2 mRNA COVID-19 Vaccine in Singapore"

_vaccines, 2022, doi:10.3390/vaccines10020331_

Round 1
Reviewer 1 Report
This study investigates the kinetics of anti-S and neutralizing Ab following the administration of a booster dose of BNT162b2 vaccine in a population originally vaccinated with the standard, two-dose, BNT162b2 vaccine. The study is well designed and the data are clearly presented. Specific points:
1) from figure 1, it seems that not all the 42 participants have been enrolled at all time points. In addition to panels A and B, showing cumulative data on all participants, it would be helpful showing longitudinal kinetics on participants enrolled at all time points, if any.
2) In the discussion, lines 129-146, authors discuss about possible protective antibody titers. In my opinion, this paragraph reports too many andecdotal observations and does not take into account one of the main factors responsible for breakthrough infections: the increased escape capability of new SARS-CoV-2 variants. I suggest shortening and rephrasing, since a protective cut-off value may not be identified due to high tendency of SARS-CoV-2 to mutate.
Author Response
REVIEWER 1
Comment 1: From figure 1, it seems that not all the 42 participants have been enrolled at all time points. In addition to panels A and B, showing cumulative data on all participants, it would be helpful showing longitudinal kinetics on participants enrolled at all time points, if any.
Reply: We only have 2 cases who underwent testing at all time points. Thus, we have included their kinetics in a supplementary figure. The antibody profile is looks similar to those in Figure 1.
Comment 2: In the discussion, lines 129-146, authors discuss about possible protective antibody titers. In my opinion, this paragraph reports too many andecdotal observations and dose not take into account one of the main factors responsible for breakthrough infections: the increased escape capability of new SARS-CoV-2 variants. I suggest shortening and rephrasing, since a protective cut-off value may not be identified due to high tendency of SARS-CoV-2 to mutate.
Reply: We have revised the discussion and added an additional references of the efficacy of booster vaccinations during the current wave of omicron infections.
Reviewer 2 Report
The manuscript by Chin Shern Lau and colleagues describes the kinetics of the antibody responses after the third dose (booster) of the Pfizer mRNA vaccine in a group of healthy volunteers in Singapore. The number of samples is limited (43 individuals, 5 time points per sample) and the group is very homogeneous: all had no evidence of viral infection (no N protein specific antibody responses) and all received 3 doses of the Pfizer vaccine. The amount of original data in the manuscript is limited to antibody titers measured by two different commercial assays. The authors also compare these new data with previous results from their own study on antibody titers after the second dose.
In general, the results are in agreement with the results being reported by other groups around the world, as the authors mention in the discussion, and confirm that: 1) the antibody titers decline after the second dose of the vaccine and 2) the third dose have a potent boosting effect. Understandably, the discussion is mainly speculative, since the correlates of immune protection for this virus (and its many variants) are still not well defined.
I believe the manuscript could be more precise if the authors addressed the following specific points:
- The results of the Snibe competitive quantitative N‐Ab assay are reported as neutralizing antibodies. Since they are not measured in a real viral neutralization assay, using infectious SARS-CoV2 virus, they authors should provide an explanation, or a reference showing the validation of this assay as a surrogate for the virus neutralization assay.
- Figure 2. The authors should indicate more clearly that the data used in this figure include data from this study (post third dose) and from their previous study (post second dose and different individuals).
- Same comment for figure 3. The line connecting the time points could be interpreted as if the points correspond to the same individuals, but samples A to G correspond to the previous study.
- In figure 4, the authors use non-linear regression to calculate the half-life of the antibodies induced after the third dose of the vaccine. If would be very interesting to compare with the results obtained after the second dose, to evaluate if the response induced by the booster is indeed longer lasting, as the authors propose in the discussion.
In the discussion, the authors speculate about the decline of the antibody titers and the possibility of breakthrough infections. I believe this section should mention and discuss the mounting evidence that some of the different variants of concern, including omicron, show reduced neutralization by the antibodies induced by the vaccines.
Author Response
REVIEWER 2
Comment 1: The results of the Snibe competitive quantitative N-Ab assay are reported as neutralizing antibodies. Since they are not measured in a real viral neutralization assay, using infectious SARS-CoV-2 virus, they authors should provide an explanation, or a reference showing the validation of this assay as a surrogate for the virus neutralization assay.
Reply: The Snibe N-Ab assay reports a clinical sensitivity of 100% (95% CI 93.69-100) in 57 samples with confirmed SARS-CoV-2 VNT50 ≥20 in their manufacturer’s insert. A recent publication has compared a plaque reduction neutralization test to the Snibe Maglumi neutralization antibody assay and have added this reference to the methods section.
Comment 2: Figure 2. The authors should indicate more clearly that the data used in this figure include data from this study (post third dose) and from their previous study (post second dose and different individuals). Same comment for figure 3. The line connecting the time points could be interpreted as if the points correspond to the same individuals, but samples A to G correspond from the previous study.
Reply: We have added the above into both figures’ explanation, and removed the connecting line between studies in figure 3.
Comment 3: In figure 4, the authors use non-linear regression to calculate the half-life of the antibodies induced after the third dose of the vaccine. If would be very interesting to compare with the results obtained after the second dose, to evaluate if the response induced by the booster is indeed longer lasting, as the authors propose in the discussion.
Reply: Based on our prior data set, we have included a non-linear regression of total and neutralizing antibodies 20-90 days post-second vaccination in a new Supplementary Figure S2 and discuss the findings in the discussion section accordingly. The half-life of the neutralizing antibody is longer after the booster vaccination.